# NTBC Treatment Monitoring in Chilean Patients with Tyrosinemia Type 1 and Its Association with Biochemical Parameters and Liver Biomarkers

**DOI:** 10.3390/jcm10245832

**Published:** 2021-12-13

**Authors:** Karen Fuenzalida, María Jesús Leal-Witt, Patricio Guerrero, Valerie Hamilton, María Florencia Salazar, Felipe Peñaloza, Carolina Arias, Verónica Cornejo

**Affiliations:** Instituto de Nutrición y Tecnología de Alimentos INTA, Universidad de Chile, Santiago 7830490, Chile; patricio.guerrero@inta.uchile.cl (P.G.); vhamilton@inta.uchile.cl (V.H.); mfsalazar@inta.uchile.cl (M.F.S.); felipe.penaloza@inta.uchile.cl (F.P.); carias@inta.uchile.cl (C.A.); vcornejo@inta.uchile.cl (V.C.)

**Keywords:** tyrosinemia type-1, nitisinone, succinylacetone, alpha fetoprotein, liver biomarkers

## Abstract

Treatment and follow-up in Hereditary Tyrosinemia type 1 (HT-1) patients require comprehensive clinical and dietary management, which involves drug therapy with NTBC and the laboratory monitoring of parameters, including NTBC levels, succinylacetone (SA), amino acids, and various biomarkers of liver and kidney function. Good adherence to treatment and optimal adjustment of the NTBC dose, according to clinical manifestations and laboratory parameters, can prevent severe liver complications such as hepatocarcinogenesis (HCC). We analyzed several laboratory parameters for 15 HT-1 patients over one year of follow-up in a cohort that included long-term NTBC-treated patients (more than 20 years), as well as short-term patients (one year). Based on this analysis, we described the overall adherence by our cohort of 70% adherence to drug and dietary treatment. A positive correlation was found between blood and plasma NTBC concentration with a conversion factor of 2.57. Nonetheless, there was no correlation of the NTBC level with SA levels, αFP, liver biomarkers, and amino acids in paired samples analysis. By separating according to the range of the NTBC concentration, we therefore determined the mean concentration of each biochemical marker, for NTBC ranges above 15–25 μmol/L. SA in urine and αFP showed mean levels within controlled parameters in our group of patients. Future studies analyzing a longer follow-up period, as well as SA determination in the blood, are encouraged to confirm the present findings.

## 1. Introduction

Hereditary tyrosinemia type-1 (HT-1) is an autosomal recessive inborn error of metabolism caused by a deficiency in the enzyme fumarylacetoacetate hydrolase (FAH), which affects the degradation pathway for tyrosine (Tyr) and leads to insufficiency in glucogenic metabolites. Globally, the prevalence of this condition is 1 per 100,000 newborns, but it has been shown to be higher in some regions—specifically, Quebec, Canada, where its prevalence is 1:1846 [1,2].

As a result of FAH dysfunction, the conversion of fumarylacetoacetate to fumarate, acetoacetate, and succinate is impaired. Accumulated 4-maleylacetoacetate and 4-fumarylacetoacetate are then reduced to succinylacetoacetate, which, in turn, is decarboxylated to succinylacetone (SA), the pathognomic biomarker of HT-1 [3]. Increased SA levels are found in biological fluids in affected newborns, and chronic elevation of these metabolites leads to the development of hepatocellular carcinoma (HCC), renal tubulopathy, glomerular disease, heart disease, and neurological problems [3,4]. Early diagnosis is crucial to promptly initiate pharmacological treatment and prevent the onset of critical complications. Treatment is based on the daily administration of 2-(2-nitro-4-trifluoromethylbenzoyl)-1,3-cyclohexanedione (NTBC), a herbicide that strongly inhibits the enzymatic activity of 4-hydroxy-phenylpyruvic dioxygenase (4HPPD), the second enzyme in the catabolic pathway of Tyr [5]. Once NTBC treatment is initiated, excretion of SA is hampered, and the levels of alpha-fetoprotein (αFP)—the liver biomarker of HCC progression—decrease slowly over a few months, until they reach normal values [6,7,8]. As the blockage caused by NTBC is in proximity to the first steps of the Tyr degradation cascade, Tyr levels increase, causing ocular problems and thrombocytopenia. To reverse these secondary complications and maintain good metabolic control, dietary restriction of phenylalanine (Phe) and Tyr is essential, in combination with NTBC treatment [9].

In Chile, 19 patients with HT-1 are under active follow-up on NTBC treatment, a Phe/Tyr restricted diet, and a Tyr-free/low-Phe protein substitute (PS) supplement. As is the case in many low- and middle-income countries, HT-1 is not included in our National Neonatal Screening Program, which only considers detection by heel stick at 40 h post-birth of Phenylketonuria and Congenital Hypothyroidism. Consequently, most of our patients were diagnosed by clinical presentation at different ages. Only one, who had a family history (siblings), was diagnosed in the neonatal period. Favorably, HT-1 patients have access to the government’s Financial Protection for High-Cost Diagnoses and Treatments System (Law No. 20850, 2015), which guarantees lifetime access to subsidized NTBC treatment and Tyr-free/low-Phe PS. Clinical, dietary, psychological, and biochemical monitoring has been carried out quarterly at our center from 1996, which is designated as the national reference center for diagnosis and follow-up of HT-1 and other IEM pathologies.

Management of HT-1 patients follows current consensus guidelines, and since 2018, our laboratory has monitored NTBC concentrations through liquid chromatography-mass spectrometry (LC–MS) in whole blood and plasma. Additionally, urinary SA levels and plasma amino acids are also routinely screened using the same methodology. NTBC dosage, dosing regimen, and the optimal therapeutic values of NTBC in the blood and plasma for management have emerged as questions in recent years [10,11,12,13]. A divided dose of NTBC has been shown to increase NTBC concentrations in the blood, compared to a once-a-day dosing system [10]. Moreover, the target range of NTBC in dried blood spot (DBS) samples needed to keep SA values at a minimum appears to be lower than the recommendation in the current consensus guidelines, as has been indicated in recently published studies [11,14].

This study aimed to associate the NTBC concentration in blood with SA levels, αFP, biochemical parameters, and liver biomarkers, while describing the overall adherence of our patients over one year of follow-up (2019–2020).

## 2. Materials and Methods

This was a retrospective observational and cross-sectional study conducted in the years 2019–2020. Fifteen HT-1 patients were included from the total of nineteen patients in active follow-up during 2019. The four patients not included in the study had been diagnosed less than a year before the study began. The clinical records of the patients between April 2019 and January 2020 provided follow-up data for one year, during which data were collected every three months (2019: April, July, October; 2020: January), as per our protocol for regular outpatient clinical visits. Following the national protocol published by the Chilean Ministry of Health [13], the entire cohort was treated as follows: NTBC recommended dose, 1 mg/kg/day (twice daily); protein restricted to 0.5 g/kg/day, supplemented with Tyr-free/low-Phe PS to 1.5–3.0 g/kg/day, considering an additional 20% to account for PS bioavailability. Infants received 400–500 mg/day, increasing in toddlers, adolescents, and adults up to 900 mg/day, depending on the Tyr and Phe plasma profile.

### 2.1. Material and Reagents

NTBC, Mesotriene, succinylacetone, and the amino acid mix were purchased from Sigma-Aldrich (Saint Louis, MO, USA) at the highest purity grade available. Internal standards were obtained from Cambridge Isotope Laboratories (Tewksbury, MA, USA., Amino acid Mix MSK-CAA-1, Labeled SU NSK-T-1). HPLC-grade solvents (methanol, acetonitrile, and water) were purchased from Merck (Saint Louis, MO, USA). Other chemical reagents, including hydrazine, formic acid (FA), and acidic butanol, were obtained from Sigma-Aldrich (Saint Louis, MO, USA). The following chromatographic columns were purchased from GLScience (Torrance, CA, USA): Symmetry C8 5 µm × 4.6 × 150 mm from Waters Company™ (Milford, MA, USA) and InertSustain C18 2 µm × 2.1 × 50 mm (Hampshire, SO51BJJ, UK).

### 2.2. Clinical Samples

DBS, plasma, and urine samples from 15 HT-1 patients were collected at each clinical visit to the center. A mean of 50 samples of each biological sample was analyzed in 2019 and 2020. For the correlation studies, paired samples from each patient at each medical check-up were analyzed as independent samples. For each biochemical parameter, the reference range for adequate adherence was DBS NTBC 20–40 µmol/L, plasma NTBC 40–60 µmol/L, SA < 0.5 mmol/mol creatinine, Tyr 400–600 µmol/L, Phe 20–80 µmol/L, and αFP < 10 µg/L [15]. The Institute of Nutrition and Food Technology (INTA) Ethics Committee granted ethical approval and approved patient-informed consent on 26 May 2021.

### 2.3. Sample Preparation for NTBC, Succinylacetone, and Amino Acids

For NTBC analysis in DBS, a 3.2 mm-diameter disk from DBS was extracted with 255 µL of methanol and 45 µL of mesotriene (10 µM). The mixture was shaken at room temperature for 60 min and then centrifuged at 11,750 rcf for 2 min. The supernatant was transferred to a 96-well ELISA plate and injected into LC-MS/MS. For plasma NTBC quantification, 30 µL of mesotriene (10 µM) and 400 µL of methanol were added to 50 µL of plasma and incubated with agitation for 60 min at room temperature. After centrifugation, 300 µL of the supernatant was transferred to a glass vial and evaporated to dryness under a stream of nitrogen. The sample was resuspended in 200 µL of 60:40 methanol/water and 0.1% formic acid and then injected into the liquid chromatography-tandem mass spectrometer (LC–MS/MS). An NTBC calibration curve and internal control were extracted and analyzed in the same run as each patient’s sample. The calibration curve corresponded to NTBC-spiked blood from a non-HT-1 affected subject (0 µM, 0.5 µM, 5 µM, 50 µM, and 100 µM).

For the SA preparation, 100 µL of filtered urine was extracted by adding 100 µL of methanol, 5 µL of SCA-C5 (0.1 mM), 20 µL water, and 375 µL hydrazine 3 mM (in 60:40 methanol/water, 0.1% FA). The mixture was incubated at 60 °C for 40 min to allow for SA derivatization. Then, 5 µL was injected into the LC–MS/MS system. Each run used an internal control and calibration curve corresponding to spiked urine from control subjects with increasing SA concentrations (0 µM, 0.5 µM, 5 µM, 50 µM, and 100 µM). Quantitative results for SA were normalized to the corresponding urinary creatinine value and are expressed as mmol SA/mol creatinine.

The amino acid sample was prepared by extracting 50 µL of plasma with 425 µL of methanol and adding 1.5 µL of labeled amino acids (MSK-CAA-1). After 10 seconds of vortexing, the vial was centrifuged at 11,750 rcf for 4 min. Then, 200 µL of the supernatant was transferred to a glass vial and evaporated to dryness under a stream of nitrogen. Butylation was carried out by resuspending with 50 µL of butanol-HCl 3N and incubating at 65 °C for 7.5 min. The sample was evaporated to dryness and resuspended in 300 µL of ACN 20% and 0.1% FA and then injected into LC–MS/MS (1 µL). The AA quantification used a linear calibration curve for each amino acid.

### 2.4. Instrumentation and Analysis by LC-MS/MS

NTBC, SA, and amino acid (AA) determinations were performed using a Shimadzu UFLC system coupled with a SCIEX 3500 ESI–MS/MS detector. For AA quantification, the chromatographic system was equipped with a C8 column (150 mm × 4.6 mm, 5 µm), while for NBTC and SA, a C18 column (50 mm × 2.1 mm × 2 µm) was used. The AAs were separated using an isocratic flow of 0.8 mL/min, with a mobile phase comprising 20% ACN and 0.1% FA kept at 40 °C. The total run was 10.5 minutes, and the sample volume injection was 10 µL. The MS/MS detector was adjusted to positive electrospray ionization at a temperature of 550 °C and source voltage of 3500 V. The ESI–MS/MS parameter was optimized for each butylated AA and its corresponding labeled standard. The analyte concentration was determined with multiple reaction monitoring (MRM) mode in order to measure the peak areas of each AA’s transition in relation to the respective internal standard peak area with a known concentration, using the MultiQuant™ 3.0.2 software. For chromatographic separation of NTBC and SA, a gradient of water/FA 0.1% (solvent A) and methanol/FA 0.1% (solvent B) was used. The total run was 6 min, the volume of sample injection for SA was 5 µL, and that for plasma NTBC was 1 µL. MS/MS detection of each analyte was through MRM in positive electrospray ionization mode. The quantifier and qualifier transitions for NTBC were 330/218 and 330/125, respectively. Furthermore, for mesotriene, it was 340/228; SA, 155/137; and SA-C5, 160/142.

Accuracy for each LC–MS/MS-based analyte determination (except plasma NTBC) was determined by analyzing external proficiency testing samples from ERNDIM (European Research Network to Evaluate and Improve Screening, Diagnosis, and Treatment of Inherited Disorders of Metabolism). The limits of quantification were as follows: SA, 0.1 µmol/L; AA, 2 µmol/L; and NTBC, 0.5 µmol/L.

### 2.5. Alpha-Fetoprotein and Liver Biomarker Determination

For HT-1 patient blood analysis of liver function, alanine transaminase (ALT), aspartate transaminase (AST), gamma-glutamyl transaminase (GGT), prothrombin time (PT), INR, bilirubin total and direct, alkaline phosphatase, and alpha-fetoprotein (αFP) were determined quarterly in the Central Laboratory at Dr. Calvo Mackenna Hospital (CMH) in Santiago, Chile. Under the Chilean program of diagnosis and follow-up of HT-1 patients, CMH and its clinical gastroenterology team collaborate on follow-up and are the designated referral center for hepatic transplantation.

### 2.6. Statistical Analysis

A descriptive analysis of the variables, including all the independent variables, was conducted using the GraphPad Prism 8.4.0 software (San Diego, CA, USA). For each variable, the distribution was determined using the Shapiro–Wilk test. To compare samples, we used the Wilcoxon test, and Spearman’s correlation was used to verify the associations between variables.

## 3. Results

Fifteen HT-1 patients in active follow-up were enrolled in the study. The mean age of patients at the time of last analysis was 11 years, 4 months (range: 1 year, 3 months to 23 years, 4 months). Two patients had less than 5 years of follow-up, and 13 patients had more than 5 years of follow-up, with a maximum of 23 years. Sixty-six percent of patients were female. Because, in Chile, there is no NBS for HT-1, 14 of the 15 patients were diagnosed by clinical manifestation; the other was diagnosed as a result of sibling diagnosis 25 days after birth. Median age at diagnosis was 7 months (range: 25 days to 3 years, 11 months). First, biological samples taken over one year at quarterly clinical and biochemical controls were analyzed as a single group in order to assess the general metabolic status of our HT-1 patients. Table 1 shows the median, mean, minimum, and maximum values for each parameter. The mean NTBC administered in a twice-daily regimen for all patients was 0.95 mg/kg/day. Pharmacological adherence to treatment was determined by measuring NTBC levels in the plasma and DBS. For plasma and DBS samples, 48% and 33% of samples were within recommended reference values [16], with means of 50.4 μmol/L and 23.6 μmol/L, respectively. Associating the effect of NTBC with SA excretion in urine, we found that more than 89% of analyzed samples presented levels below 0.5 mmol/mol of creatinine, even though 30% of analyzed samples showed lower-than-recommended NTBC levels in plasma (<40 μmol/L). Dietary adherence during the period was determined by comparing the Tyr and Phe levels with the reference range described in the management protocol, under which the Tyr and Phe levels should be kept between 400 and 600 μmol/L and 20 and 80 μmol/ L, respectively. We observed that 67% of the analyzed samples were within the recommended Tyr range, with an average value of 584.2 μmol/L (range: 319.4–1181.6), and 83% were within the recommended Phe range (mean: 47.8 μmol/L; range: 18.1–76.7). It is worth noting that 60% of patients with HT-1 were on Phe supplementation.

### 3.1. Comparison of NTBC Concentration in Plasma and DBS

To date, few studies have addressed whether the NTBC concentrations measured in plasma and DBS are correlated, and whether the conversion factor is equivalent in different HT-1 patient cohorts [17,18,19]. In a paired study of 43 plasma and DBS samples, we identified the correlation coefficient for NTBC concentrations between the two types of samples. We found a significant correlation between NTBC levels measured in DBS and plasma (Spearman r: 0.8046; *p* < 0.0001), with a conversion factor of 2.57 (Figure 1), which we can use to transform NTBC values from DBS into plasma values. Given our determination of equivalence between the two types of samples, and considering the most accepted range for NTBC levels in plasma samples (40–60 μmol/L), the calculated range for NTBC in DBS would be 15–24.5 μmol/L (38.6–64 μmol/L in plasma).

### 3.2. Association of Succinylacetone Excretion in Urine with NTBC Concentration

Table 2 shows the mean and median NTBC concentrations for both plasma and DBS in a set of samples grouped according to SA excretion levels found in urine samples taken at the same time as the plasma and DBS. The first cut-off value (COV) was 0.25 mmol/mol creatinine; we established a second at 0.5 mmol/mol creatinine to evaluate whether significant differences in mean NTBC could be found, in relation to the SA excretion level. Nearly 80% of the analyzed urine samples presented SA values below 0.25 mmol/mol creatinine. When the recommended limit was raised to 0.5 mmol/mol creatinine, the percentage increased to 89.7%. In both sets of samples, slight or no significant differences were found in mean NTBC concentration, whether detected in plasma or DBS, with SA limits at 0.25 and 0.5 mmol/mol creatinine. The mean NTBC concentration in DBS was 22.9 and 23 μmol/L in samples with SA below 0.25 and 0.5 mmol/mol creatinine, respectively, and 14.4 and 15.5 μmol/L, respectively, in samples with SA levels higher than the above mentioned COV.

### 3.3. Determination of Urinary Levels of Succinylacetone and Alpha-Fetoprotein According to NTBC Concentration Range

Paired samples were sorted by NTBC concentration (in DBS): 0–14.9 μmol/L, 15–24.9 μmol/L, 25–34.9 μmol/L, and >35 μmol/L. For each range, we determined the mean concentration of SA in urine and the hepatocarcinogenic biomarker αFP (Figure 2 and Table 3). Recommended optimal management values for SA and αFP were identified for the NTBC range in DBS from 15–24.9 μmol/L (SA below 0.5 mmol/mol creatinine and αFP below 10 μg/L). The mean for SA was 0.2 mmol/mol creatinine (95% CI range: 0–0.4) and 8.3 μg/L for αFP (95% CI range: 4.8–10.6). Levels close to 5 μg/L of αFP (5.5) were found only at NTBC concentrations higher than 35 μmol/L. No significant correlation was found between NTBC and SA or αFP in the paired-sample comparison.

### 3.4. Association of Plasma Amino Acids with NTBC Concentration

We correlated each laboratory monitoring variable with the NTBC concentration in DBS for all samples taken at the four clinical controls in the year of follow-up. No significant correlation was found between NTBC levels and Tyr, Phe, or Phe–Tyr ratios. Interestingly, a significant positive correlation was found when methionine concentrations were analyzed with NTBC level (Spearman r = 0.2963; *p* = 0.0239). The Phe–Tyr restricted diet had good adherence in our patients; however, 30% of analyzed samples registered higher-than-recommended levels. Thus, by segregating NTBC concentrations, as above, we evaluated whether higher NTBC levels significantly influenced amino acid levels in our cohort of patients. Figure 3 shows that the median Tyr levels (Figure 3A) were within the recommended range (400–600 μmol/L) when NTBC concentrations in DBS were up to 34.9 μmol/L. However, in samples with higher NTBC levels (>35 μmol/L), elevated Tyr median values outside the acceptable reference range were observed (no statistical difference was found for lower NTBC ranges). In contrast, the Phe concentration remained within optimal values for all the NTBC ranges (Figure 3B). Methionine levels showed an upward trend, which was dependent on NTBC concentration but remained within good metabolic control limits (Figure 3C).

### 3.5. Association of Liver Biomarkers with NTBC Concentrations

At clinical appointments, we also monitored liver biomarkers. Table 4 shows the median values for each parameter measured in the study, including coagulation factor (prothrombin time), transaminases (ALT, AST, GGT), direct and total bilirubin, and alkaline phosphatase. Overall, the median values were within the recommended reference values, except for alkaline phosphatase, for which the median value was more than twice the maximum recommended limit, and GGT, for which levels remained outside the normal reference range.

NTBC association with liver biomarkers was addressed through correlation analysis. No significant correlation was found between NTBC concentration in DBS and any liver biomarker listed in Table 4. Liver parameters over one year of follow-up in HT-1 patients did not reveal any significant trends or difference based on NTBC levels segregated by range (Appendix A). Alkaline phosphatase was higher than recommended for all ranges of NTBC concentrations.

## 4. Discussion

During a one-year follow-up period prior to COVID-19 outbreak, we retrospectively observed the overall status of each laboratory parameter and compared them to recommended reference values. Then, we established a conversion factor for NTBC concentration in plasma and DBS samples, allowing us to carry out future NTBC monitoring on DBS, improving the follow-up of HT-1 patients affected by socio-economical and geographical limitations. Finally, we determined the mean values of principal HT-1 follow-up biomarkers in different ranges of NTBC concentration in blood samples.

In this study, a general analysis of adherence to drug treatment for the period observed revealed that 35% of the tested samples were within the recommended range for the plasma NTBC concentration [16]; however, when all the samples with values greater than 30 μmol/L were included, the percentage increased to 86%. Plasmatic NTBC concentration above 30 μmol/L has been shown to be sufficient to maintain SA levels below 0.25 mmol/mol creatinine in urine, as has recently been demonstrated [13]. Eighty percent of patients received an NTBC dose between 0.66–1.2 mg/kg/day, and two patients (>15 years old) were treated with doses of 0.6 mg/kg/day, and still presented plasma NTBC values >30 μmol/L with SA excretion values below 0.5 mmol/mol creatinine. Pharmacological treatment is administered as two doses/day, which may explain the optimal concentration of NTBC in plasma, which has been shown to be beneficial in maintaining stable NTBC levels in the blood and effectively preventing an increase in SA [10,14,20]. However, this may negatively impact adherence to treatment, leading to poorer metabolic control [21,22]. In our analysis, we found that 70% of HT-1 patients presented with adequate nutritional adherence, achieved by a controlled diet restricted in Phe/Tyr. We are aware that more in-depth and long-term studies are needed in order to better describe dietary and pharmacological adherence in this cohort.

Upon measuring the NTBC concentration in plasma and DBS samples, we found a slight difference in the conversion factor (cf) calculated for NTBC concentrations in plasma and DBS, compared to the inter-laboratory studies carried out by Laeremans [17] (cf = 2.4) and the Spanish group (cf = 2.34) [18]. Our determination was slightly higher (2.57) and consistent with our initial reports in 2019 [19]. The ethnicity of our population, the wide age range (1 year, 3 months to 23 years, 4 months), and the age-dependent hematocrit value could explain the difference between our findings and those of the European cohorts. It is important to note that our LC–MSMS method for the NTBC quantification was first validated by measuring NTBC levels in DBS samples from Chilean HT-1 patients in parallel with the U.K. Center at Evelina Children’s Hospital [19].

The SA level in HT-1 patients should be managed below the limits of detection or within limits established by each laboratory [17,23,24], which is dependent on instrument sensitivity [17]. We established a first COV for urine SA monitoring of 0.5 mmol/mol creatinine. For this COV, 89% of samples presented levels below 0.5 mmol/mol creatinine. Interestingly, when a lower COV was used (0.25 mmol/mol creatinine), no significant difference in median NTBC levels was observed regarding to COV 0.5 mmol/mol creatinine. These results suggest no major difference in NTBC concentrations for a COV of less than 0.5 mmol/mol creatinine for SA. On the other hand, monitoring of SA in DBS has been strongly encouraged and validated in the past few years [11,14]. In a recently published study, the average NTBC concentration in blood in paired samples with SA levels < 0.3 μmol/L in DBS was 17.2 μmol/L [14], lower than the mean value found in our study (21.2 μmol/L), which considered a COV of SA in urine of 0.5 mmol/mol creatinine. These differences can be explained by the sensitivity of the method, daily variation of SA levels, the stability of the matrix, and other factors. Future comparative studies of SA stability and simultaneous determination of its concentration in plasma, blood, and urine are required. Despite such differences, our findings are consistent with Yeo [14] and Shultz’s [11] observations that target NTBC levels in the blood could be lower than currently recommended. Nonetheless, this primary finding should be considered carefully, and in light of the limitations of our study, further prospective studies with a greater number of samples and longer observational times are encouraged to be performed in the future. Additionally, standardized SA measurement in blood samples will allow us to unify criteria and contribute to evaluating future clinical recommendations regarding to the values of blood NTBC for therapy monitoring.

Periodic evaluation of other liver biomarkers has been proposed as essential for closely monitoring the outcome of hepatic complications [7]. Although no significant differences could be found between the association of NTBC levels with SA levels in urine and αFP—the primary marker for HCC—we observed a negative association trend. Moreover, with a 95% confidence interval, we observed that, above the range of 15–24.9 μmol/L of NTBC in DBS, SA levels and αFP fell within the recommended monitoring parameters [16,25,26,27]. Although a COV of 10 μg/L for αFP has been established in several reviews and consensus guidelines [28], subtle elevation within this range might indicate a progression in liver damage, especially in clinically diagnosed patients. No other robust biomarkers have yet been found to support αFP in predicting the risk of carcinogenesis. An example of this is a patient included in this study with late diagnosis (12 months) who developed, during the observed period, an abrupt elevation of αFP (from 7.3 μg/L to 2470 μg/L within 3 months), concomitant with evidence of suppressed levels of SA (<0.2 mmol/mol creatinine). This illustrates the need to search for better biomarkers to predict prognosis, particularly considering that patients with late diagnosis, such as ours, will probably have greater liver complications during their lifetimes [12,22,28].

Beyond the αFP analyzed herein, no major associations between transaminase levels, coagulation measurements, or bilirubin and NTBC concentrations could be found. However, some parameters of liver function, such as alkaline phosphatase and GGT, were elevated in relation to our laboratory’s normal reference range. It has been reported that NTBC slightly elevates transaminase values [29], which was supported by our observations. The association analysis of NTBC with liver biomarkers in a heterogeneous cohort of HT-1 patients that included a majority of patients with more than 5 years of NTBC treatment (13 from 15 patients) suggested no greater variability on biomarkers levels related to NTBC blood concentration in this one-year observational study. In-depth studies are required to clarify the exact impact of other variables on chronic liver damage in patients with late diagnoses, such as long-term exposure to NTBC.

We did not find any significant association of NTBC with Tyr and Phe, unlike that which has been recently reported by Gonzalez-Lamuño [30], who found a positive correlation with Tyr levels and the Phe/Tyr ratio. This difference can be explained by the poor dietary adherence of Spanish HT-1 patients, where 54%–64.4% of patients exceeded recommended Tyr levels. In contrast to the Spanish cohort, we found 70% adherence to dietary treatment, and Tyr levels tended to be outside acceptable management values only at the highest range of NTBC concentrations (>35 μmol/L). On the other hand, methionine was positively correlated with NTBC, but the levels were within acceptable follow-up ranges even at high concentrations of NTBC. The role of methionine in chronic liver disease has recently been reviewed by Li, Z. et al. [31] and warrants consideration as an indicator of progressive liver dysfunction in HT-1.

## 5. Conclusions

This is the first study carried out in Chile that has attempted to associate NTBC levels with biochemical parameters in HT-1 patients who are part of the 24-year-old follow-up tyrosinemia program. Our results revealed that our cohort of HT-1 patients presented acceptable overall compliance with NTBC and dietary treatment, despite diversity in age, time exposure to NTBC, and clinical onset. Additionally, we demonstrated that measurement of NTBC in DBS is well correlated with NTBC plasma levels, enabling us include them in laboratory monitoring in the follow-up program. Data extracted from a series of HT-1 patients with a clinical diagnosis (not by neonatal screening) with long-term follow-up not only reaffirmed the evidence obtained in countries that have high economic resources, but also suggests the utility of medium- and low-income countries to engage in follow-up of their patients according to international guidelines and publish their experiences, beyond the difficulties they may have in achieving adequate diagnosis and treatment for their patients.

## Figures and Tables

**Figure 1 jcm-10-05832-f001:**
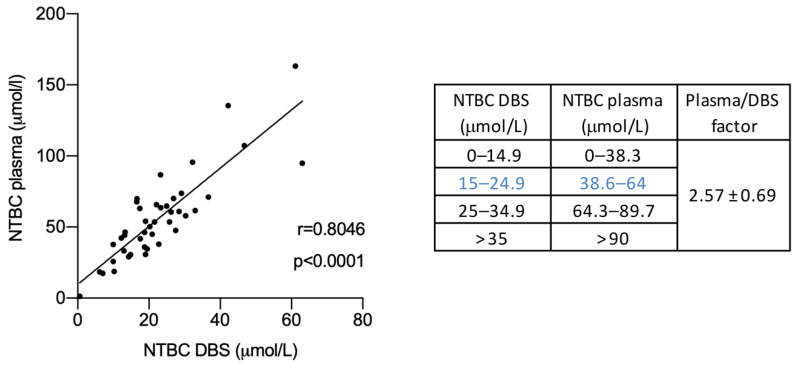
Correlation of the NTBC concentration for 43 plasma (umol/L) and DBS (μmol/L) samples (*r* = 0.8046; *p* < 0.001; 95%CI 0.74–0.92). The complement table on the right shows the conversion values for each range with a factor of 2.57 ± 0.69. DBS, Dried blood spot.

**Figure 2 jcm-10-05832-f002:**
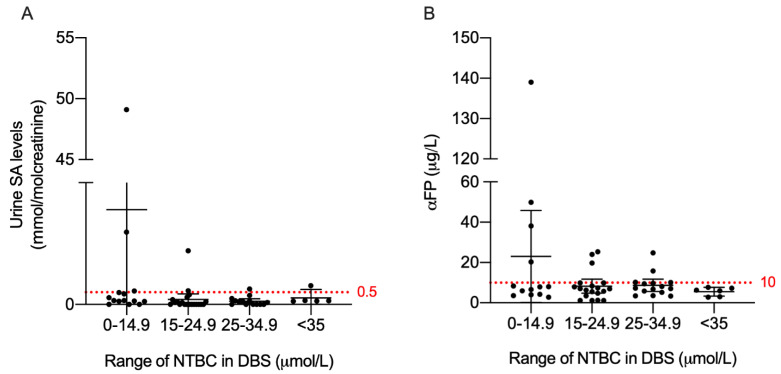
NTBC concentration ranges associated with SA excretion in urine and αFP in plasma. Graph (**A**): SA; Graph (**B**): αFP. Dotted lines represent the maximum allowable concentration: SA, 0.5 mmol/mol creatinine in urine; αFP, 10 μg/L in plasma. NTBC, Nitisinone; DBS, Dried blood Spot; SA, Succinylacetone; αFP, alpha-fetoprotein.

**Figure 3 jcm-10-05832-f003:**
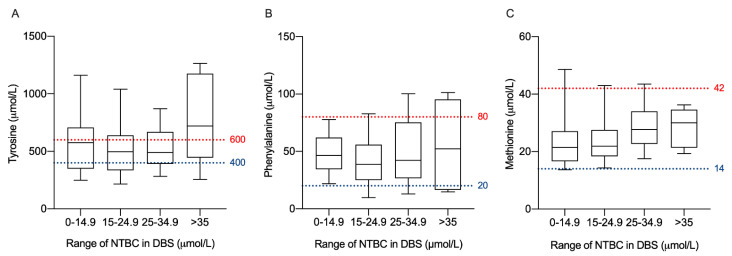
Amino acid variation according to NTBC concentration ranges. Boxplots of Tyr (**A**); Phe (**B**); and Met (**C**) by NTBC concentration ranges. Each boxplot represents the 25th, 50th, and 75th quartiles for each amino acid. Dotted lines represent maximum and minimum allowable concentrations under clinical guidelines: Tyr, 400–600 μmol/L (for patients >6 years old); Phe, 20–80 μmol/L; Met, 14–42 μmol/L. Tyr, tyrosine; Phe, phenylalanine; Met, Methionine; DBS, dried blood spot.

**Table 1 jcm-10-05832-t001:** Laboratory follow-up parameters for Chilean patients with HT-1.

Parameter	Analysis	Value	Samples (*n*)	Recommended Reference Value *
NTBC doses (mg/kg/day)	Median	0.97	57	1.0 mg/kg/day
	Mean ± SD	0.95 ± 0.17		
	Min–max	0.66–1.31		
NTBC concentration (μmol/L) DBS	Median	21.3	57	20–40 μmol/L
	Mean ± SD	23.6 ± 12.6		
	Min–max	3.46–60.14		
NTBC concentration (μmol/L) Plasma	Median	50.6	43	40–60 μmol/L
	Mean ± SD	50.4 ± 21.8		
	Min–max	9.8–101.6		
Tyrosine plasma (μmol/L)	Median	460.7	58	400–600 μmol/L
	Mean ± SD	584.2 ± 253.4		
	Min–max	319.4–1181.6		
Phenylalanine plasma (μmol/L)	Median	47.4	58	20–80 μmol/L
	Mean ± SD	47.8 ± 19.5		
	Min–max	18.05–76.7		
Methionine plasma (μmol/L)	Median	23.83	58	14–43 μmol/L
	Mean ± SD	25.18 ± 6.4		
	Min–max	16.5–40.5		
Succinylacetone (mmol/mol creatinine)	<0.5	89%	53	<0.5 mmol/mol creatinine
	>0.5	11%		

* According to consensus guidelines: Chinsky et al., 2017 [16]; Van Sprosen, 2017 [9].

**Table 2 jcm-10-05832-t002:** NTBC concentration in plasma and DBS samples at two COVs for succinylacetone levels in urine.

		NTBC DBS				NTBC Plasma			
	% of Total Samples	Mean	Number of Samples	Median	*p*-Value	Mean	Number of Samples	Median	*p*-Value
Samples with SA < 0.25 mmol/mol creatinine	79.5	22.9	31	22.1	* 0.027	57.9	31	60.5	* 0.025
Samples with SA > 0.25 mmol/mol creatinine	20.5	14.4	8	14.8		35.8	8	42.1	
Samples with SA < 0.5 mmol/mol creatinine	89.7	23	35	21.2	0.323	55.1	35	53.5	0.357
Samples with SA > 0.5 mmol/mol creatinine	10.3	15.5	4	15.6		38.8	4	48.2	

* *p*-value < 0.05, according to the Wilcoxon test. NTBC, Nitisinone; DBS, dried blood spot.

**Table 3 jcm-10-05832-t003:** The mean of SA and αFP levels according to the NTBC concentration range in blood samples.

Range of NTBC in DBS (µmol/L)	Number of Paired Samples	SA in Urine (mmol/mol Creatinine)	Number of Paired Samples	αFP (µg/L)
0–14.9	14	Mean ± SD: 3.9 ± 13.1; CI 95%: 0–11	13	Mean ± SD: 23 ± 37.8; CI 95%: 0.1–44.5
15–24.9	19	Mean ± SD: 0.2 ± 0.5; CI 95%: 0–0.41	19	Mean ± SD: 8.3 ± 7.2; CI 95%: 3.5–10.6
25–34.9	14	Mean ± SD: 0.1 ± 0.2; CI 95%: 0–0.22	15	Mean ± SD: 8.7 ± 5.5; CI 95%: 4.8–10.8
<35	5	Mean ± SD: 0.2 ± 0.3; CI 95%: 0–0.46	6	Mean ± SD: 5.5 ± 2.1; CI 95%: 2.3–6.8

**Table 4 jcm-10-05832-t004:** Liver parameters over one year of follow-up in HT-1 patients.

		Values	Samples Analyzed (*n*)	Recommended Reference Value
Prothrombin time (sec)	Median	13.7	48	11–13.5 sec
	Mean ± SD	14.6 ± 4.1		
	Min–max	7.45–31		
INR	Median	1.06	48	0.8–1.1
	Mean ± SD	1.09 ± 0.12		
	Min–max	1.0–1.8		
Aspartate aminotransferase AST (UI/L)	Median	35.5	52	15–40 UI/L
	Mean ± SD	37.9 ± 18.5		
	Min–max	13–103		
Alanine aminotransferase ALT (UI/L)	Median	25	52	10–34 UI/L
	Mean ± SD	30.2 ± 23		
	Min–max	10–154		
ALT/AST	Median	0.72	52	1
	Mean ± SD	0.79 ± 0.28		
	Min–max	0.37–1.56		
Gamma-glutamyl transferase (GGT)	Median	27.5	52	11–21 UI/L
	Mean ± SD	38 ± 34		
	Min–max	10–202		
Bilirubin direct (mg/dL)	Median	0.13	48	0.3 (mg/dL)
	Mean ± SD	0.19 ± 0.24		
	Min–max	0.02–1.72		
Bilirubin total (mg/dL)	Median	0.39	48	0.2–1.2 (mg/dL)
	Mean ± SD	0.48 ± 0.31		
	Min–max	0.13–1.78		
Alkaline Phosphatase (UI/L)	Median	204.5	45	44–147 (UI/L) *
	Mean ± SD	262 ± 149		
	Min–max	62–976		

* Age- and gender-dependent range. INR, international normalized ratio; sec, seconds.

## Data Availability

The data can be obtained by contacting the first author, K.F.

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
