# Peer review of "NTBC Treatment Monitoring in Chilean Patients with Tyrosinemia Type 1 and Its Association with Biochemical Parameters and Liver Biomarkers"

_jcm, 2021, doi:10.3390/jcm10245832_

Round 1

Reviewer 1 Report

This manuscript reports observations in 15 patients with HT1 comparing NTBC dosing and dietary treatment with NTBC concentrations in dried blood spots and plasma, succinylacetone excretion in urine, alpha fetoprotein in blood, plasma amino acids and biochemical markers like transaminases, GGT, bilirubin and ALP in blood. Based on a one-year follow up assessment the authors intend to describe overall adherence to treatment and the optimal NTBC dose. After reading this very lengthy manuscript and considering time of the study and parameters analyzed/presented this goal is not achieved. One year follow up is not sufficient to determine whether a specific NTBC dose influences liver integrity or function. Furthermore, using urine succinylacetone measurement to determine NTBC dose effects on suppression of succinylacetone formation is not sensitive. Instead, one would have to use plasma or DBS succinylacetone. In its present form, the paper does not add new information to the body of knowledge.

Many aspects of the manuscript need correction with respect to terminology and content. Please note, newborn screening programs nowadays include succinylacetone as a first tear test. That allowed to increase diagnostic sensitivity and specificity for HT1 screening. Considering that the majority of NBS samples is taken on day two of life such success would not been possible if the authors were correct with their statement that ‘increased SA levels are found in biological fluids as early as 48 hrs’.

Author Response

  1. “One year follow up is not sufficient to determine whether a specific NTBC dose influences liver integrity or function”.

Our analysis aimed to address whether there is an association between NTBC concentration and biochemical/liver parameters levels using data over one-year (2019-2020) from HT-1 patients who are periodically controlled in our follow-up program. So, the present study included data from long-term NTBC treated patients (up to 20 years) as well as short-term ones (one-year). Assessment of liver function or integrity in NTBC-treated patients or under a long-term exposure to NTBC was not considered one of the scopes of our research. We agree that one year follow-up is not sufficient, and long-term evaluation data as well as a larger cohort of HT-1 subjects are required to achieve such association.  

2.- “Using urine succinylacetone measurement to determine NTBC dose effects on suppression of succinylacetone formation is not sensitive”.

Certainly, the measurement of succinylacetone (SA) in urine is less sensitive and reproducible than SA determination in plasma or whole blood. However, the use of urine SA for the monitoring of NTBC-dependent suppression of SA synthesis is supported by the current international guidelines and expert recommendations (Chinsky et al, 2017; van Ginkel WG, 2019). Furthermore, therapeutic range validation of NTBC by using SA in urine has been recently published by Jack and Scott (Jack, 2019). Sensitivity of SA detection in plasma or DBS may also be variable among laboratories, which is dependent on instrument capability, and the establishment of limit of detection/quantification as shown by Laeremans, 2020.

It is worth noting that, in this retrospective data set, we only had access to evaluate SA in urine and NTBC in plasma and DBS as established in the current governmental health protocol for HT-1 disease. Considering that monitoring of SA in DBS has been strongly encouraged and validated in recent years given its advantages over urine and plasma samples, we are working on a proposal for the Health Ministry that considers the determination of SA in DBS in the National HT-1 follow-up protocol.

4.- Their statement that ‘increased SA levels are found in biological fluids as early as 48 hrs”.

This inadvertent imprecision was corrected in the introduction (page1, line 36). It is possible that the meaning of this affirmation was not entirely clear. In Chile, as in many low- and middle- income countries, the NBS of HT-1 is not included as part of the health governmental program. Almost all HT-1 diagnoses are by clinical diagnosis, not before 48 hr of life. We included in page 2, line 170, the following sentence: “As is the case in many low- and middle-income countries, HT-1 is not included in our National Neonatal Screening Program, which only considers detection, by heel stick at 40 hours of life of Phenylketonuria and Congenital Hypothyroidism.”

Other considerations: This manuscript was submitted for English edition by MDPI's regular publishing service. We condensed the discussion to shorten the length of the manuscript. Terminology and some clarifications were also included as was directly indicated by reviewer# 3.

References:

  1. Chinsky JM, Singh R, Ficicioglu C, van Karnebeek CDM, Grompe M, Mitchell G, et al. Diagnosis and treatment of tyrosinemia type I: a US and Canadian consensus group review and recommendations. Genet Med. 2017;19:1380–1380. https://doi.org/10.1038/gim.2017.101
  2. van Ginkel WG, Rodenburg IL, Harding CO, Hollak CEM, Heiner-Fokkema MR, van Spronsen FJ. Long-Term Outcomes and Practical Considerations in the Pharmacological Management of Tyrosinemia Type 1. Pediatr Drugs. 2019;21:413–26. https://doi.org/10.1007/s40272-019-00364-4
  3. Jack RM, Scott CR. Validation of a therapeutic range for nitisinone in patients treated for tyrosinemia type 1 based on reduction of succinylacetone excretion. JIMD Rep. 2019;46:75–8. https://doi.org/10.1002/jmd2.12023
  4. Laeremans H, Turner C, Andersson T, Juan JAC, Gerrard A, Heiner‐Fokkema MR, et al. Inter‐laboratory analytical improvement of succinylacetone and nitisinone quantification from dried blood spot samples. JIMD Rep. 2020;53:90–102. https://doi.org/10.1002/jmd2.12112

Reviewer 2 Report

This is a well written paper that adds clinical insights to the management of patients affected by tyrosinemia type 1, showing that lower NTBC concentration in blood might be sufficient to control levels of SA and αFP, leading to possible suggestions of adjusting the dose to an effective minimum.

Author Response

No minor or major corrections were suggested by this reviewer.

Reviewer 3 Report

Though this paper does not contain many new findings it is interesting to summarize the experience in Chile which by and large similar to the experience in other countries.

There are a number of spelling and grammatical errors. Thorough revision is required.

p.1, l33: Tyr does not always accumulate in the untreated patient, please rephrase

p.1, l 44: alpha-fetoprotein drops very slowly, this should be mentioned

p. 3, l 130: 12000rpm; this should be expressed as x g

The discussion is too long and should be shortened

Author Response

1.- -“There are a number of spelling and grammatical errors. Thorough revision is required”.

Manuscript was edited by MDPI English editing services (regular edit option that checks grammar, spelling, punctuation, and phrasing of the paper). Hereby, the attached edition certificate can be found.

2.- “p.1, l33: Tyr does not always accumulate in the untreated patient, please rephrase.”

Effectively, this was an imprecision that we corrected as followed (p.1, l33):

As a result of FAH dysfunction, the conversion of fumarylacetoacetate to fumarate, acetoacetate, and succinate is impaired. The accumulated 4-maleylacetoacetate and 4-fumarylacetoacetate are then reduced to succinylacetoacetate which, in turn, is decarboxylated to succinylacetone (SA), the pathognomic biomarker of HT-1 [3].

3.- “p.1, l 44: alpha-fetoprotein drops very slowly, this should be mentioned” 

In this phrase, we included the followed sentence (p.1, l 46):

Once NTBC treatment is initiated, excretion of SA is hampered, and the levels of alpha-fetoprotein (αFP)—the liver biomarker of HCC progression—decreases slowly over a few months, until it reaches normal values

4.- “p. 3, l 130: 12000rpm; this should be expressed as x g”

Rpm unit was changed to rcf (xg) in two sections:

p.3, l 275  “then centrifuged at 11,750 rcf for 2 min”

p.3, l 296   “was centrifuged at 11,750 rcf for 4 min”

5.- “The discussion is too long and should be shortened”.

Discussion was condensed and shortened from 2031 to 1317 words. Second paragraph was eliminated, and the extensive discussion of particular results was either summarized or removed.

Round 2

Reviewer 1 Report

The merit of this manuscript lies in the comparison between NTBC measurements in plasma and DBS. The attempt to use the observations in this one-year study to draw conclusions on recommended NTBC dosing lacks experimental evidence and must be excluded from the manuscript. The same applies for attempts the study the association of NTBC level and liver markers and NTBC level and Tyrosine concentrations. The former lacks sensitivity due to the short observation period, only highly toxic effects would be visible. The latter is cofounded by diet adherence. Succinylacetone quantification in urine is not sensitive enough to determine the therapeutic dose of NTBC independent of the method that is applied. The only valid approach is the correlation of NTBC dosing/ measured NTBC concentrations with blood succinylacetone quantification. To avoid misguidance of clinicians the authors need to omit any conclusions on optimal NTBC dosing.

Author Response

We appreciate you the time dedicated to the revision of our manuscript and for the insight comment and observations.  We are confident that your recommendations will improve the scientific quality of our paper. We are aware that some sentences in the previous manuscript could mislead the clinical recommendation for the management of patients with tyrosinemia. To correct this point, we have incorporated changes in the statements throughout the manuscript that avoid stating the clinical recommendation of a specific dose of NTBC. In addition, we eliminated those conclusions in which we had established the optimal NTBC concentration range based on SA determination in urine.

We have focused the approach of the manuscript and the objective on a more descriptive analysis of the state of our cohort, which has the particularity of being very heterogeneous in terms of the time that patients have been under NTBC and nutritional treatment follow-up (patients with one year of treatment, the majority with more than 5 years and others with more than 20 years). Even so, the general adherence during a year of observation with four complete clinical and biochemical appointments showed positive results. This observation was consistent with the smallest fluctuations observed in Tyrosine levels when we analyzed by different ranges of NTBC concentration in blood, except for one that had higher NTBC concentrations, which may be a reflect of the difficulty in maintaining the adherence to a stricter nutritional diet. This point was one we intended to address with the association analysis of amino acids according to the NTBC concentration range.

Additionally, we included in the discussion section a sentence with the present limitations of our study and what aspects would be crucial to be addressed in the future, for example, association analysis of SA levels in blood with NTBC concentration and prospective studies of the biomarkers with longer observational time.

We have marked the changes in the document with text editor and sent this version for English editing.